# L-Type Amino Acid Transporter 1 Regulates Cancer Stemness and the Expression of Programmed Cell Death 1 Ligand 1 in Lung Cancer Cells

**DOI:** 10.3390/ijms222010955

**Published:** 2021-10-11

**Authors:** Yi-Heng Liu, Yu-Ling Li, Huan-Ting Shen, Peng-Ju Chien, Gwo-Tarng Sheu, Bing-Yen Wang, Wen-Wei Chang

**Affiliations:** 1Department of Pulmonary Medicine, Taichung Tzu Chi Hospital, Buddhist Tzu Chi Medical Foundation, Taichung 427, Taiwan; tc1921301@tzuchi.com.tw (Y.-H.L.); ryenhat@tzuchi.com.tw (H.-T.S.); 2Department of Biomedical Sciences, Chung Shan Medical University, No. 110, Sec. 1, Jianguo N. Rd., Taichung City 40201, Taiwan; ss021342225@gmail.com (Y.-L.L.); chienpengju@gmail.com (P.-J.C.); 3Division of Thoracic Surgery, Department of Surgery, Changhua Christian Hospital, No. 135 Nanhsiao Str., Changhua City 50006, Taiwan; 4Institute of Medicine, Chung Shan Medical University, No. 110, Sec. 1, Jianguo N. Rd., Taichung City 40201, Taiwan; gtsheu@csmu.edu.tw; 5School of Medicine, Chung Shan Medical University, No. 110, Sec. 1, Jianguo N. Rd., Taichung City 40201, Taiwan; 6School of Medicine, College of Medicine, Kaohsiung Medical University, No. 100, Shih-Chuan 1st Road, Sanmin Dist., Kaohsiung City 80708, Taiwan; 7Institute of Genomics and Bioinformatics, National Chung Hsing University, No. 145 Xingda Rd., South Dist., Taichung City 40227, Taiwan; 8College of Medicine, National Chung Hsing University, No. 145 Xingda Rd., South Dist., Taichung City 40227, Taiwan; 9Ph.D. Program in Translational Medicine, National Chung Hsing University, No. 145 Xingda Rd., South Dist., Taichung City 40227, Taiwan; 10Department of Medical Research, Chung Shan Medical University Hospital, No. 110, Sec. 1, Jianguo N. Rd., Taichung City 40201, Taiwan

**Keywords:** LAT1, PD-L1, cancer stem cells, non-small cell lung cancer

## Abstract

The l-type amino acid transporter 1 (LAT1) is a membranous transporter that transports neutral amino acids for cells and is dysregulated in various types of cancer. Here, we first observed increased LAT1 expression in pemetrexed-resistant non-small cell lung cancer (NSCLC) cells with high cancer stem cell (CSC) activity, and its mRNA expression level was associated with shorter overall survival in the lung adenocarcinoma dataset of the Cancer Genome Atlas database. The inhibition of LAT1 by a small molecule inhibitor, JPH203, or by RNA interference led to a significant reduction in tumorsphere formation and the downregulation of several cancer stemness genes in NSCLC cells through decreased AKT serine/threonine kinase (AKT)/mammalian target of rapamycin (mTOR) activation. The treatment of the cell-permeable leucine derivative promoted AKT/mTOR phosphorylation and reversed the inhibitory effect of JPH203 in the reduction of CSC activity in pemetrexed-resistant lung cancer cells. Furthermore, we observed that LAT1 silencing caused the downregulation of programmed cell death 1 ligand 1 (PD-L1) on lung cancer cells. The PD-L1^+^/LAT1^+^ subpopulation of NSCLC cells displayed great CSC activity with increased expression of several cancer stemness genes. These data suggest that LAT1 inhibitors can serve as anti-CSC agents and could be used in combination with immune checkpoint inhibitors in lung cancer therapy.

## 1. Introduction

Lung cancer, the second most common cancer in both sexes combined, is the leading cause of death among cancers all over the world. Approximately 85% of lung cancers are non-small cell lung carcinoma (NSCLC), of which lung adenocarcinoma and lung squamous cell carcinoma are the most common subtypes. In addition to the traditional treatments for NSCLC, immune checkpoint inhibitors (ICIs), including therapeutic antibodies to programmed cell death protein 1 (PD-1) or programmed cell death ligand-1 (PD-L1), display therapeutic breakthroughs in NSCLC, but the objective response rate (ORR) is not high [1]. In a study with pretreated older NSCLC patients, the ORR was only 27.4% for ICI monotherapy [2]. The development of combination agents for ICIs has been suggested to enhance the ORR. One potential strategy is to inhibit the key molecules that regulate the expression level of immune checkpoint molecules [3]. For example, the combination of erlotinib, an epidermal growth factor receptor (EGFR) tyrosine kinase inhibitor (TKI), with nivolumab, an anti-PD-1 antibody, has been evaluated clinically in advanced NSCLC patients with EGFR mutations [4]. However, some adverse effects were observed in the combination of EGFR-TKI and PD-1 blockade therapy [5]. Thus, the search for other potential combination agents for ICIs is still required.

Cancer stem cells (CSCs), also known as tumor-initiating cells, are a subpopulation of cancer cells within tumors that play the key role in tumor initiation, growth, metastasis, resistance to therapies, and relapse [6]. Targeting CSCs has been considered as a potential strategy for efficient cancer therapy [7]. It has been reported that PD-L1 expression could be enriched in CSCs, which contributes to the immune evasion of CSCs by suppressing T cell immunity [8]. In addition, PD-L1 expression on NSCLC cells could transduce signals for proliferation and cell invasion through the upregulation of β-catenin through the phosphoinositide 3-kinase (PI3K)/Akt pathway [9]. However, the involvement of PD-L1 in CSC maintenance is not fully understood.

An increased uptake of essential amino acids (EAAs) has been found in tumor tissues when compared to the adjacent normal parts [10], which is mediated by amino acid transporters including the l-type amino acid transporter (LAT) family. LATs consist of four Na^+^-independent neutral amino acid transporters (LAT1-4) which function as an important route for EAAs’ entry into cells [11]. LAT1 (gene name SLC7A5) consists of 12 transmembrane domains and covalently binds to the heavy chain of 4F2 antigen (gene name SLC3A2) to form a heterodimeric functional complex and is mainly responsible for the influx of large branched-chain and aromatic neutral amino acids into cells, such as leucine [12]. The overexpression of LAT1 has been found in a variety of cancers. High expression of SLC7A5 mRNA and LAT1 protein was associated with poor prognosis in the luminal B type of breast cancer [13]. Recently, LAT1 has been reported to participate in the chemoresistance of docetaxel in luminal type breast cancer cells [14]. Nevertheless, the information about the changes of LAT1 expression level in pemetrexed-resistant NSCLC cells is still limited. In H69 small cell lung cancer cells, knockdown of LAT1 by short interfering RNA (siRNA) inhibited cell proliferation by inducing G1 cell cycle arrest [15]. Recently, Lu et al. found that the elevated expression of LAT1 was associated with poor overall survival of NSCLC patients [16]. However, the functional role of LAT1 expression in the CSC maintenance, as well as in the expression of PD-L1, in NSCLC cells remains unclear.

In the current study, we investigated the difference of LAT1 expression between pemetrexed-sensitive and resistant NSCLC cells and the involvement of LAT1 in CSC activity of NSCLC cell lines. In addition, we further identified the positive regulatory function of LAT1 to PD-L1 expression on NSCLC cells and discovered the potential use of LAT1+PD-L1^+^ as a CSC marker for NSCLC. Our data suggest the applicative potential of LAT1 inhibitors in combination with ICIs for future NSCLC therapy.

## 2. Results

### 2.1. LAT1 Is Elevated in Pemetrexed-Resistant NSCLC Cells

We had previously established pemetrexed-resistant NSCLC cells from A549 cells by continued treatment with 400 nM of pemetrexed; these cells are called A400 cells [17]. The CSC activity of A400 cells was found to have increased when compared to parental A549 cells using tumorsphere cultivation (Figure 1A). The mRNA expression of several cancer stemness genes including BMI1, Sox2, Oct4, and c-Myc, was upregulated in A400 cells (Figure 1B). We previously found that the sensitivity of H1299 cells to pemetrexed was between A549 cells and A400 cells [17]. Thus, the H1299 cells were included to investigate the relationship between LAT1 and pemetrexed sensitivity. With quantitative reverse transcription-polymerase chain reaction (qRT-PCR) and fluorescence-activated cell sorter (FACS) analyses, we found the mRNA expression of SLC7A5 (Figure 1B) and the protein expression of LAT1 on the cell surface (Figure 1C) had increased in A400 cells, although the total protein expression of LAT1 in H1299 cells was less than that of A549 cells (Appendix A). We further found that the LAT1 protein expression level was negatively correlated with pemetrexed sensitivity after comparison between A549, A400, and H1299 cells (Figure 1D). The lung adenocarcinoma data of the Cancer Genome Atlas (TCGA) database displayed the significant variations in LAT1 mRNA expression between clinical stages (Figure 1E) and was significantly correlated with shorter overall survival (Figure 1F). These data suggest that LAT1 may regulate the maintenance of the CSCs in NSCLC cells.

### 2.2. The Inhibition of LAT1 Suppresses CSC Activity in NSCLC Cells

We next examined the role of LAT1 in CSC activity of NSCLC cells. When using lentiviral delivery of LAT1-specific shRNA into A400 cells or transfection of LAT1-specific siRNA into H1299 cells to knockdown LAT1 expression, the formation of tumorspheres was suppressed (Figure 2A). In addition, the knockdown of LAT1 in A400 and H1299 cells downregulated several cancer stemness genes including BMI1, Sox2, and Oct4 (Figure 2B). It is known that the transport of amino acids by LAT1 in cancer cells could activate the mTOR pathway [19]. We also found the decreased phosphorylation of Akt and mTOR in LAT1 knockdown A400 and H1299 cells (Figure 2C). In addition to RNA interference, we also used JPH203, the small molecule inhibitor of LAT1 [20], to inhibit LAT1 activity. The formation of tumorspheres in A400 and H1299 cells was decreased by JPH203 treatment in a dose-dependent manner (Figure 3A). The downregulation of Akt or mTOR phosphorylation was also observed in JPH203-treated A400 or H1299 cells (Figure 3B). In addition, the treatment of JPH203 caused the downregulation of BMI1 or SOX2 (Figure 3C), two of the stemness genes which have been demonstrated to participate in CSC maintenance in NSCLC [21]. To further demonstrate the requirement of amino acid transport activity of LAT1 in CSC activity, the cell permeable leucine analog, l-Leucyl-l-Leucine methyl ester (LLME), was added in combination with JPH203. The results revealed that the addition of LLME recovered the tumorsphere forming capability (Figure 3D) as well as induced the phosphorylation of Akt and mTOR (Figure 3E) in both A400 and H1299 cells. With the forced expression of myristoylated Akt (myr-Akt), a constitutively active form of Akt [22], the inhibitory activity of JPH203 at a concentration of 10 μM in tumorsphere formation was reduced, but not for the concentration at 25 μM (Figure 3E). These data suggest that LAT1 participates in CSC activity by activating the Akt/mTOR pathway partially and contributing to the expression of stemness genes in others.

### 2.3. Inhibiting the Expression or Activity of LAT1 Reduces PD-L1 Expression on NSCLC Cells

Lastwika et al. previously demonstrated that the Akt/mTOR pathway could drive PD-L1 expression on NSCLC cells [23]. In addition, the link between PD-L1 and CSC-like cells has also been reported [24]. Due to the participation of LAT1 in Akt/mTOR activation and in the self-renewal capability being observed in our data (Figure 2 and Figure 3), we decided to determine if LAT1 involves PD-L1 expression on NSCLC cells. The knockdown of LAT1 by RNA interference was found to reduce PD-L1 expression in terms of total protein levels (Figure 4A) and on the cell surface (Figure 4B) of A400 and H1299 cells by Western blot and fluorescence-activated cell sorting (FACS) analysis, respectively. Inhibition of LAT1 activation by JPH203 treatment also obtained similar results of RNA interference (Figure 4C,D). These data suggest that LAT1 activity is involved in PD-L1 expression on NSCLC cells.

### 2.4. Silencing the PD-L1 Expression Decreases CSC Activity in NSCLC Cells

Almozyan et al. have reported that tumoral PD-L1 positively regulates Oct4 or Nanog expression to maintain CSC activity in breast cancer [25]. We next examined the involvement of PD-L1 in cancer stemness of NSCLC cells. Using the lentiviral delivery of PD-L1-specific shRNA followed by examining CSC activity by tumorsphere cultivation, the knockdown of PD-L1 significantly reduced tumorsphere formation in A549 and H1299 cells (Figure 5A). To explore the underlying molecular mechanisms in PD-L1 regulating CSC maintenance, the RNA sequencing technique and GSEA analysis were applied and the data revealed that the knockdown of PD-L1 in A549 cells downregulated the pathways involving G2/M checkpoints, E2F1 targets, TNF-α signaling via NF-κB, hypoxia, or EGFR upregulated signatures with the criteria of a nominal *p* value less than 0.05 and a false discovery *q*-value less than 0.25 according to the suggestions from the GSEA website (Figure 5B). The enriched gene sets of E2F1 targets were also found among subjects with PD-L1 mRNA expression higher than the median level and were also found in NSCLC patients of the TCGA database (Appendix A). The expression of several genes involved in these gene sets were confirmed by quantitative RT-PCR, and the results revealed that the knockdown of PD-L1 in A549 cells reduced the expression of EZH2, JAG1, IGFBP3, CAV, TGFB2, WEE1, TYMS, KLF4, and CDC25A but increased the expression of PDCD4 and GPX3, two potential tumor suppressors in lung cancers [26,27] (Figure 5C).

### 2.5. Coexpression of LAT1 and PD-L1 Enriches the CSC Population in NSCLC Cells

Due to the observations of the positive roles of LAT1 or PD-L1 expression in CSC activity in NSCLC cells, we next used the parental A549 or H1299 cells, rather than the pemetrexed-resistant A400 cells, to understand the general impact of the expression patterns of cell surface LAT1/PD-L1 on the CSC activity of NSCLC cells. The cell sorting was performed after incubation of fluorescent-conjugated anti-LAT1 and/or anti-PD-L1 antibodies (Figure 6A) and was followed by the examination of CSC activity by tumorsphere cultivation. The results revealed that LAT1^+^/PD-L1^+^ cells displayed higher tumorsphere forming capabilities than those of LAT1^−^PDL1^−^ or LAT1^−^/PD-L1^+^ cells derived from A549 or H1299 cells (Figure 6B). The LAT1^+^/PD-L1^+^ A549 cells displayed the greatest expression levels of EZH2, BMI1, c-Myc, and Oct4 proteins (Figure 6C). After analyzing the TCGA database via the GEPIA2 webtool, we found that although a single gene of CD274 did not show any significant correlation to the overall survival among NSCLC patients, the two-gene signature of CD274 and SLC7A5 displayed a significantly positive correlation with shorter overall survival of NSCLC patients (Figure 6D). These data indicate that LAT1^+^/PD-L1^+^ functions as a novel CSC marker for NSCLC cells.

## 3. Discussion

LAT1 is an antiporter that imports neutral amino acids simultaneously with the exportation of glutamine [28]. Leone et al. previously demonstrated that a glutamine blockade by JHU083, a small molecule glutamine antagonist through the inhibition of glutaminase activity, in tumor-bearing mice led to the reduction of hypoxia and the promotion of acidosis of tumor cells, but T cell activation was induced within the tumor microenvironment [29]. Our data demonstrated that the inhibition of LAT1 activity led to the downregulation of PD-L1 on NSCLC cells (Figure 4). Using the GEPIA2 webtool to analyze an NSCLC dataset in the TCGA database, we also found that SLC7A5 mRNA was significantly negatively correlated with the gene signatures of Th1-like, effector T-cells, and resident memory T-cells (Appendix A); this lead to a hypothesis that LAT1 expression levels in NSCLC tumors may contribute to the immunosuppressive tumor microenvironment. In addition, Yazawa et al. discovered that the coexpression of LAT1 and ASC amino acid transporter 2, the glutamine transporter that mediates the cellular uptake of glutamine, serves as an independent prognostic factor of NSCLC patients [30]. It is also known that the uptake of kynurenine, the tryptophan metabolite produced by indoleamine 2,3-dioxygenase or tryptophan-2,3-dioxygenase, by natural killer (NK) cells or T cells within the tumor microenvironment is mediated by LAT1, which leads to the suppression of their proliferation and effector functions [31]. These observations also suggest that the application of glutamine antagonists and LAT1 inhibitors may overcome the immune suppressive tumor microenvironment through the downregulation of PD-L1 or reduce the uptake of immunosuppressive amino acid metabolites into NK or T cells.

Although some amino acid deprivations, including glutamine and asparagine depletion [32], have been suggested as effective strategies in cancer therapy, opposite outcomes have been observed for leucine deprivation in cancer cells. Xiao et al. found that the deprivation of leucine induced apoptosis of breast cancer cells through the decreased expression of fatty acid synthase [33]. In contrast, Viana et al. demonstrated that a leucine-rich diet caused the metabolic shift of tumor cells from glycolysis to oxidative phosphorylation and led to the reduction of metastasis in a Walker-256 rat tumor model [34]. However, the amino acid starvation culture or the knockdown of LAT1 in EGFR expressing tumor cells could induce necroptosis to enhance the cytotoxic effect of gefitinib, an EGFR tyrosine kinase inhibitor [35]. It suggests that the combination of chemotherapeutic agents with amino acid deprivation has the clinical potential for cancer therapy. In the results of the overexpressing myr-Akt in NSCLC cells, it only overcame the suppressive effect of JPH203 in tumorsphere formation at the low concentration (Figure 3F), which indicates other pathways are involved in LAT1-mediated CSC maintenance in NSCLC. Dann et al. previously reported that the lung cancer cells with a high expression level of LAT1 displayed the high import of methionine, which led to the elevated activation of EZH2 [36]. Furthermore, the expression level of LAT1 and EZH2 was correlated with a less differentiated state in the tumorspheres of lung cancer cells [36]. It is possible that the activity of EZH2 also involves the LAT1-mediated CSC maintenance but this needs to be investigated in the future.

Yu et al. demonstrated that PD-L1 expression promotes tumor growth and progression in lung cancer through activating the β-catenin pathway [9]. In our study, we further demonstrated that the expression of PD-L1 positively regulated the maintenance of CSCs in NSCLC cells (Figure 5) and the expression of LAT1^+^/PD-L1^+^ could serve as a novel marker for NSCLC-CSCs (Figure 6). We also observed that the two-gene signature of SLC7A5 and CD274 displayed positive correlations with EZH2, c-Myc, and SOX2 in the NSCLC dataset of the TCGA database (Appendix A), which further supports the conjecture that LAT1^+^/PD-L1^+^ is a CSC marker in NSCLC cells. It has been proposed that the tumor intrinsic expression of PD-L1 on NSCLC cells may function as a tumor suppressor [37]. However, the addition of an anti-PD-1 or anti-PD-L1 antibody in NSCLC cells induced the activation of Akt or ERK in vitro and increased tumor growth in vivo [37]. It has been reported that the activation of the Akt/mTOR pathway could lead to the activation of STAT3 to participate in the maintenance of cancer stemness of NSCLC cells [38]. These previous reports suggest that anti-PD-1 or anti-PD-L1 antibody therapy may induce CSC activity in NSCLC, which could possibly lead to secondary ICI resistance. It also brings a possibility that the combination of anti-CSC agents with ICIs might serve as an effective strategy for overcoming ICI resistance.

## 4. Materials and Methods

### 4.1. Cell Lines

A549 and H1299 NSCLC cells were obtained from Prof. Jiunn- Liang Ko (Institute of Medicine, Chung Shan Medical University, Taichung, Taiwan), which were originally purchased from American Type Culture Collection (Manassas, VA, USA). Cells were cultured in Dulbecco′s Modified Eagle Medium (DMEM, Gibco, Thermo Fisher Scientific, Waltham, MA, USA) and supplemented with 10% fetal bovine serum, 1 mM sodium pyruvate, 2 mM l-glutamine, 100 μg/mL penicillin/streptomycin/amphotericin B and non-essential amino acids in a humidified incubator at 37 °C with 5% CO_2_. A400 cells, the pemetrexed-resistant NSCLC cells derived from A549 cells as in the previous report [39], were cultured according to the above conditions except for the addition of pemetrexed at a concentration of 400 nM to maintain the resistance.

### 4.2. Tumorsphere Cultivation

Cells were suspended in DMEM/F12 medium containing 0.5% methylcellulose (Sigma-Aldrich, St. Louis, MO, USA), 4% bovine serum albumin (fraction V, Hyclone Laboratories, Inc., Logan, UT, USA), 10 ng/mL EGF (PeproTech Asia, Rehovot, Israel), 10 ng/mL bFGF (Sino Biological Inc., Beijing, China), 2.5 µg/mL insulin (Sigma-Aldrich), 0.5 X B27 supplement (Gibco), 1 μg/mL hydrocortisone (Sigma-Aldrich), and 4 μg/mL heparin (Sigma-Aldrich) and seeded into an ultralow attachment 6-well-plate (Greiner Bio-One, Kremsmünster, Austria) at a density of 5000 cells/well (for primary tumorspheres) or 2000 cells/well (for secondary tumorspheres). For the experiments in the knockdown of PD-L1 in A549 cells, the initial seeding cell number was 1 × 10^4^ cells/well. The formed tumorspheres were counted and pictured under an inverted light microscope.

### 4.3. Pharmacological Agents

JPH203 (Cat# SML1892) was purchased from Sigma-Aldrich. l-Leucyl-l-Leucine methyl ester (hydrochloride) (Cat# 6491-83-4) was purchased from Cayman Chemical (Ann Arbor, MI, USA). In the experiments, JPH203 and l-Leucyl-l-Leucine methyl ester (hydrochloride) were dissolved in dimethyl sulfoxide (DMSO, Mallinckrodt Baker, Inc., Phillipsburg, NJ, USA).

### 4.4. RNA Interference

The RNA interference was performed by lentiviral delivery of gene-specific short hairpin RNA (shRNA) or by the transfection of gene-specific siRNA oligonucleotides. For lentiviral delivery of shRNAs, vectors carrying human LAT1 specific shRNAs (shSLC7A5-1, TRCN0000333467; shSLC7A5-2, TRCN0000333531), human PD-L1 specific shRNAs (sh-PD-L1#1, TRCN0000056915; sh-PD-L1#2, TRCN0000056916) or LacZ specific shRNA (TRCN0000231722) were obtained from the National RNAi Core Facility at the Institute of Molecular Biology (Academia Sinica, Taipei City, Taiwan). To generate the lentiviruses, HEK 293T cells were transfected with packaging plasmid (pCMV–△R8.91), envelope (pMD.G), and short hairpin pLKO-RNAi vectors using a T-pro NTRII transfection kit (T-Pro Biotechnology, New Taipei City, Taiwan). At 16 h after transfection, the culture media were replaced with fresh DMEM/F-12 medium containing 1% BSA to collect the virus fluids for a further 48 h. For virus transduction, cells were seeded at 1 × 10^5^ cells per well in 6-well plates and transduced with lentivirus in the presence of 8 μg/mL polybrene for 36 h. The transduced cells were selected with 2 μg/mL puromycin for 48 h. For siRNA transfection, LAT1 siRNA (Cat. No. sc-62555, Santa Cruz Biotechnology, Inc., Dallas, TX, USA) or control siRNA (Cat. No. sc-37007) were first diluted in Opti-MEM I Reduced Serum Medium (Gibco) at a final concentration of 100 nM and complexed with 7.5 µL TransIT-X2^TM^ transfection reagent (Mirus Bio LCC., Madison, WI, USA) at room temperature for 30 min. The TransIT-X2:DNA complexes were then added into wells. At 48 h after transfection, the transfected cells were harvested for further experiments. The efficiency of knockdown was subsequently determined by RT-PCR or Western blotting.

### 4.5. Western Blot Analysis

Total cellular proteins were isolated in NETN lysis buffer (150 mM NaCl, 20 mM Tris-HCl, pH 8.0, 0.5% NP-40, 1 mM EDTA) supplemented with protease inhibitor cocktail (Merck Millipore, Middlesex County, MA, USA) and phosphatase inhibitor cocktail (Merck Millipore, Germany). The total protein concentration in each sample was then determined with a BCA Protein Assay Kit (ThermoFisher Scientific, Inc., Waltham, MA USA). Thirty µg of extracted protein were resolved in 8–10% SDS-PAGE and transferred to the PVDF membrane (Pall Corporation, Port Washington, NY, USA). After being incubated with 5% skim milk in Tris-buffered saline −0.1% Tween 20 (TBS-T) for 1 h at room temperature to block non-specific binding, the membrane was incubated with primary antibodies at 4 °C overnight and secondary antibodies conjugated with horseradish peroxidase (HRP) at room temperature for 1 hr. The signals were then developed by incubation with chemiluminescence substrate (PerkinElmer Inc., Waltham, MA, USA) and captured with a Luminescence-Image Analyzer (FUSION SOLO, Vilber Lourmat Deutschland GmbH, Germany). The primary antibodies that were used were LAT1 (Cat# sc-374232, 1:1000; Santa Cruz Biotechnology), p-mTOR (Ser 2481) (Cat# sc-293132, 1:1000; Santa Cruz Biotechnology), mTOR (Cat# gtx101557b, 1:1000; GeneTex International Corporation, Hsinchu City, Taiwan), p-Akt1/2/3 (Ser 473) (Cat# sc-135651, 1:1000; Santa Cruz Biotechnology), Akt (Cat# 4685, 1:1000; Cell Signaling Technology, Inc., Danvers, MA, USA), GAPDH (Cat# gtx100118, 1:10,000; GeneTex), and β-Actin (Cat# A5441, 1:5000; Sigma-Aldrich). Secondary antibodies used were Anti-rabbit IgG, HRP-linked Antibody (1:10,000; Jackson ImmunoResearch, West Grove, PA, USA) and Anti-mouse IgG, HRP-linked Antibody (1:10,000; Jackson ImmunoResearch). The original images of western blot data were shown in the Appendix A).

### 4.6. Plasmid DNA Transfection

The cells were seeded in 6-well plates (5 × 10^4^ cells/well) and incubated at 37 °C overnight. The culture medium was replaced with fresh DMEM medium containing 2% FBS, and the cells were transfected with 2 µg of a myr-Akt expression vector (a gift from Professor Che-Hsin Lee, Department of Biological Sciences, National Sun Yat-sen University, Kaoshiung City, Taiwan) or a negative control vector of jetPrime transfection reagent (Polyplus transfection, Sébastien Brant, Illkirch, France). At 24 h after transfection, the transfected cells were harvested for further experiments.

### 4.7. RNA Extraction, RNA Sequencing, and Real-Time Reverse Transcription-Polymerase Chain Reaction (RT-PCR)

Total RNA was extracted and purified by a Quick-RNA MiniPrep Kit (Zymo Research, Irvine, CA, USA). For RNA Sequencing analysis, 3 μg of isolated total RNA were qualified and sequenced by Biotools Biotech Co. Ltd. (Taipei City, Taiwan). The raw reads of RNA fragments were normalized using the RLE, TMM, or FPKM method. The Gene Set Enrichment Analysis (GSEA) was performed by the GSEA software (version 4.1.0, Broad Institute, Inc., Cambridge, MA, USA). For real-time RT-PCR, 1 μg total RNA was reverse-transcribed to cDNA by an RNA RevertAid First Strand cDNA Synthesis Kit (Thermo Fisher Scientific, USA). RT-PCR was performed by using SYBR Green Master Mix (Bio-Rad Laboratories, Hercules, CA, USA) and an Eco 48 real time PCR system (PCR max, Staffordshire, UK) with specific qPCR primer pairs whose sequences are listed in Appendix A. The expression levels of each gene were normalized to mRPL19 in the same sample, and the 2^−ΔΔCT^ method was implemented to analyze the results.

### 4.8. FACS Analysis

The PerCP conjugated mouse monoclonal anti-LAT1 antibody (clone name: BU53) was purchased from Novus Biologicals, LLC. (Centennial, CO, USA). The PE- or APC-conjugated mouse monoclonal anti-PD-L1 antibodies (clone name: 10F.9G2) were purchased from Biolegend (San Diego, CA, USA). For analysis of cell surface expression of LAT1 or PD-L1, A549 or H1299 cells were harvested by enzyme-free dissociation solution (Gibco), resuspended in FACS buffer (0.4% BSA in PBS buffer containing 0.05% NaN3), and stained with fluorescence-conjugated antibodies on ice for 30 min. The fluorescence signals were detected with FACS Canto-II flow cytometry (BD Biosciences, Franklin Lakes, NJ, USA). For isolation of NSCLC cells with different LAT1 or PD-L1 expression statuses, A549 or H1355 cells were stained with PE-conjugated anti-PD-L1 antibody and PerCP-conjugated anti-LAT1 antibody at room temperature for 1 h. The LAT1^−^/PD-L1^−^, LAT1^−^/PD-L1^+^, or LAT1^+^/PD-L1^+^ cells were then sorted by a FACSAria II cell sorter (BD Biosciences).

### 4.9. Statistical Analysis

Quantitative data were presented as the mean ± SD. The comparisons between two groups were analyzed with the Student′s *t*-test; the comparisons among multiple groups (more than two) were analyzed with ANOVA. Two tailed *p*-values less than 0.05 were considered statistically significant for all tests.

## 5. Conclusions

We demonstrated that the upregulation of LAT1 in pemetrexed-resistant NSCLC cells is associated with increased CSC activity and the inhibition of LAT1 by RNA interference or JPH203 suppressing the self-renewal capability, which is mediated through the Akt/mTOR pathway. In addition, we discovered that LAT1 activity contributes to surface PD-L1 expression on NSCLC cells and identified that PD-L1 expression contributes to the CSC activity of NSCLC cells. Furthermore, NSCLC cells with LAT1^+^/PD-L1^+^ markers displayed a great CSC activity. In summary, our observations suggest that LAT1 targeting agents can potentially be developed as a combination drug for ICIs, and the CSC targeting strategies may overcome the ICI resistance in NSCLCs.

## Figures and Tables

**Figure 1 ijms-22-10955-f001:**
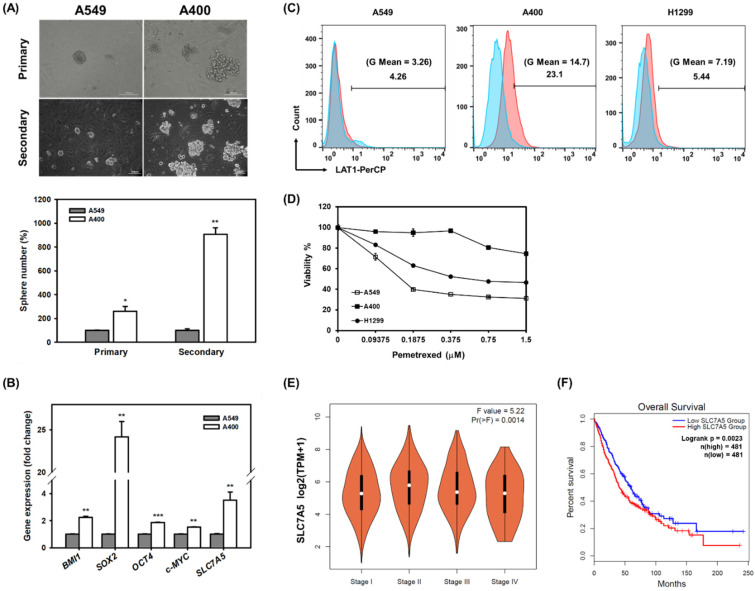
LAT1 expression increases in pemetrexed-resistant NSCLC cells. (**A**) The primary or secondary tumorspheres were cultured from A549 and A400 cells according to the protocol described in the Materials and Methods section. Pictures were presented at 200× magnification. The inserted scale bars indicate 100 μm in length. Quantitative data were presented as the related percentage to the tumorsphere number of A549 cells. * *p*< 0.05; ** *p*< 0.01. (**B**) The mRNA expressions of cancer stemness genes (BMI1/SOX2/Oct4/c-Myc) and SLC7A5 in A549 and A400 cells were determined by real-time RT-PCR. ** *p*< 0.01; *** *p*< 0.001. (**C**) The membrane expression of LAT1 was determined by FACS. Blue peaks and red peaks represent an isotype control antibody and an anti-LAT1 antibody, respectively. (**D**) The pemetrexed sensitivity was determined at 96 h after treatment and read out by MTT reagent. (**E**) The correlation of SLC7A5 mRNA expression with stages of NSCLC patients in the TCGA database was analyzed using the GEPIA2 webtool [18]. TPM, transcripts per million. (**F**) The overall survival curves of NSCLC patients in the TCGA database based on SLC7A5 mRNA expression levels were obtained from the GEPIA2 webtool using median expression level as the cutoff criteria.

**Figure 2 ijms-22-10955-f002:**
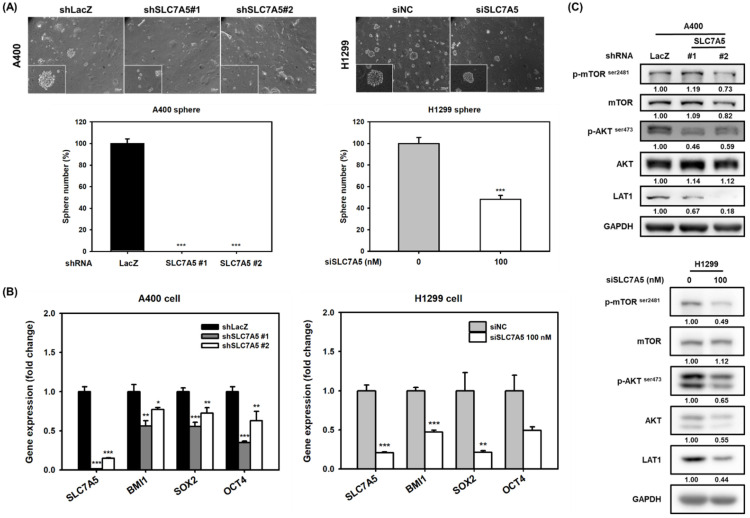
Knockdown of LAT1 reduces CSC activity and Akt/mTOR activation in NSCLC cells. (**A**) The knockdown of LAT1 in A400 and H1299 cells was performed by lentiviral delivery of LAT1-specific shRNAs (shSLC7A5#1 or shSLC7A5#2) and 100 nM of LAT1-specific siRNA oligonucleotides, respectively, for 48 h. The CSC activity was determined by tumorsphere cultivation. The inserted scale bars indicate 100 μm in length. *** *p*< 0.001. (**B**) The mRNA expressions of cancer stemness genes (BMI1/SOX2/Oct4) and SLC7A5 were determined by real-time RT-PCR. * *p* < 0.05; ** *p* < 0.01; *** *p* < 0.001. (**C**) The protein expressions of phosphor-mTOR^ser2481^, mTOR, phosphor-Akt^ser473^, Akt, and LAT1 were determined by Western blot analysis. GAPDH was used as a protein-loading control. The inserted numbers present relative expression folds in comparison to shLacZ or the control siRNA group.

**Figure 3 ijms-22-10955-f003:**
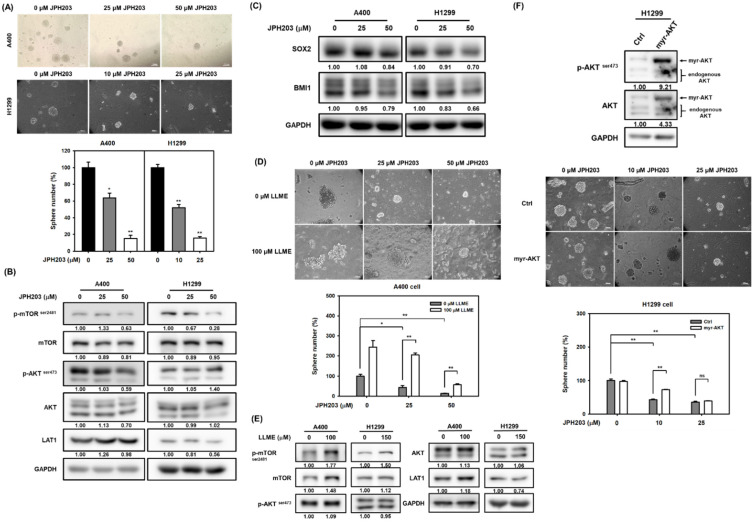
Inhibition of LAT1 activity by JPH203 suppresses CSC activity and Akt/mTOR activation in NSCLC cells. (**A**) The CSC activity of A400 and H1299 cells was determined by tumorsphere cultivation, and pictures were presented at 100× magnification. The quantitative data were presented as relative percentages to the tumorsphere number of the 0.1% DMSO treated group (0 μM). The inserted scale bars indicate 100 μm in length * *p* < 0.05; ** *p* < 0.01. (**B**) The protein expressions of phosphor-mTOR^ser2481^, mTOR, phosphor-Akt^ser473^, Akt, and LAT1 were determined by Western blot analysis. GAPDH was used as a protein-loading control. The inserted numbers presented are relative expression folds in comparison to the 0.1% DMSO treated group (0 μM) after normalization with the internal control of GAPDH. (**C**) The protein expression of BMI1 or SOX2 was determined by Western blot. The inserted numbers indicated relative expression levels as compared to DMSO group after normalization to the internal control of GAPDH. (**D**) The CSC activity of A400 cells under JPH203 treatment with or without 100 μM LLME cotreatment was determined by tumorsphere cultivation and pictures were presented at 200× magnification. The quantitative data were presented as relative percentages of the tumorsphere number of the 0.1% DMSO without LLME treatment group. The inserted scale bars indicate 100 μm in length. * *p* < 0.05; ** *p* < 0.01. (**E**) The protein expressions of phosphor-mTOR^ser2481^, mTOR, phosphor-Akt^ser473^, Akt, and LAT1 in A400 and H1299 cells in the presence of LLME were determined by Western blot analysis. GAPDH was used as a protein-loading control. The inserted numbers are presented as relative expression folds in comparison to the cells without LLME treatment group after normalization to GAPDH. (**F**) The CSC activity of H1299 cells under JPH203 treatment with or without the overexpression of the myr-Akt vector was determined by tumorsphere cultivation, and pictures were presented at 100× magnification. The inserted scale bars indicate 100 μm in length. The quantitative data were presented as relative percentages of the tumorsphere number of the control vector transfected and 0.1% DMSO treatment group. The overexpression of myr-Akt (upper panel) was determined by Western blot analysis. ** *p* < 0.01. ns, not significant.

**Figure 4 ijms-22-10955-f004:**
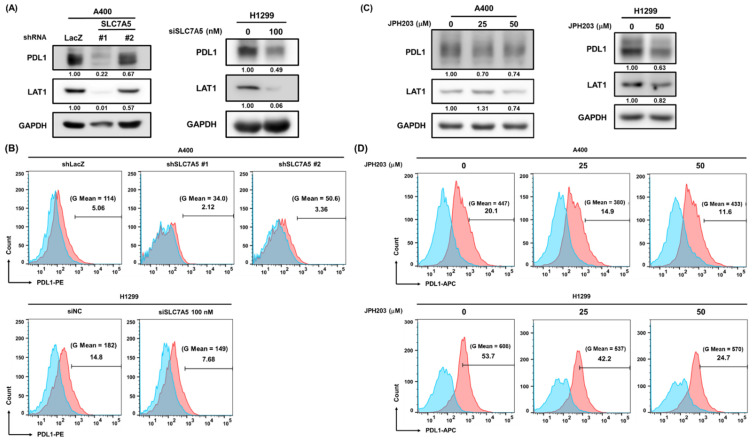
LAT1 activity participates in PD-L1 expression on NSCLC cells. (**A**,**B**). The knockdown of LAT1 by lentiviral delivery of LAT1-specific shRNAs (shSLC7A5#1 or shSLC7A5#2) for A400 cells or by transfection of LAT1-specific siRNA oligonucleotides (si-SLC7A5) for H1299 cells was confirmed by Western blot analysis (**A**). The cell surface expressions of PD-L1 were determined by Western blot (**A**) or FACS analysis (**B**), respectively. The FACS data was analyzed by FlowJo software. The inserted numbers in (**A**) represent the relative expression levels of PD-L1 or LAT1 in comparison to the sh-LacZ or si-NC (indicated as 0 nM) control group, respectively, after normalization with GAPDH. Blue peaks and red peaks in (**B**) represent an isotype control and an anti-PD-L1 antibody, respectively. siNC, negative control siRNA. (**C**,**D**) The PD-L1 total protein and cell surface expressions of A400 and H1299 cells under JPH203 treatment were determined by Western blot (**C**) and FACS analysis (**D**), respectively. The inserted numbers in (**A**) present the relative expression levels of PD-L1 or LAT1 in comparison to the 0.1% DMSO control group (0 μM) after normalization with GAPDH. Blue peaks and red peaks in (**D**) represent an isotype control and an anti-PD-L1 antibody, respectively.

**Figure 5 ijms-22-10955-f005:**
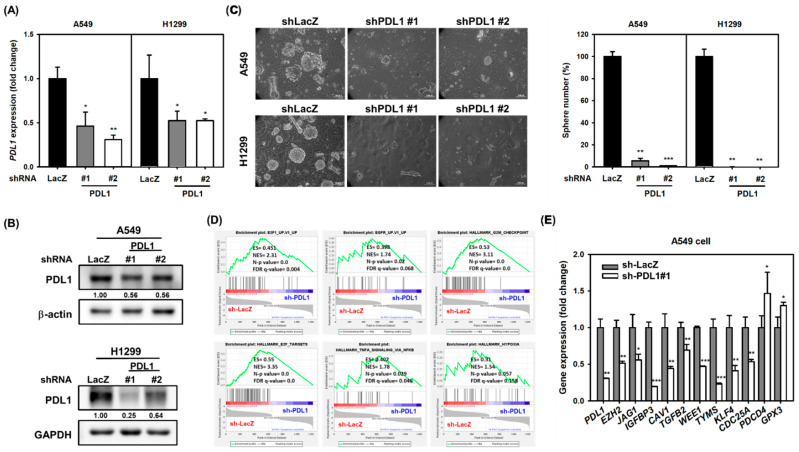
Knockdown of PD-L1 inhibits CSC activity in NSCLC cells. (**A**–**C**) The knockdown of PD-L1 in A549 or H1299 cells was performed by lentiviral delivery of PD-L1-specific shRNAs (shPDL1#1 or shPDL1#2). The mRNA (**A**) or protein expression (**B**) of PD-L1 were determined by qRT-PCR or Western blot analysis, respectively. The inserted numbers in (**B**) indicated the relative expression levels in comparison to sh-LacZ group after normalization to GAPDH. * *p* < 0.05; ** *p* < 0.01. The CSC activity was determined by tumorsphere cultivation (**C**). The inserted scale bars indicate 100 μm in length. Data were presented as relative percentages of the tumorsphere number of the shLacZ group. ** *p* < 0.01; *** *p* < 0.001. (**D**) The GSEA analysis of RNA sequencing data from shLacZ or shPDL1#1 lentivirus transduced A549 cells was performed by GSEA software. ES, enrichment score; NES, normalized enrichment score; *n*-*p* value, nominal *p* value; FDR *q*-value, false discovery rate *q*-value. (**E**) The mRNA expressions of selected genes in A549 after transduction of shLacZ or shPDL1#1 lentiviruses were determined by real-time RT-PCR. Data were presented as relative expression levels to the shLacZ group. * *p* < 0.05; ** *p* < 0.01; *** *p* < 0.001.

**Figure 6 ijms-22-10955-f006:**
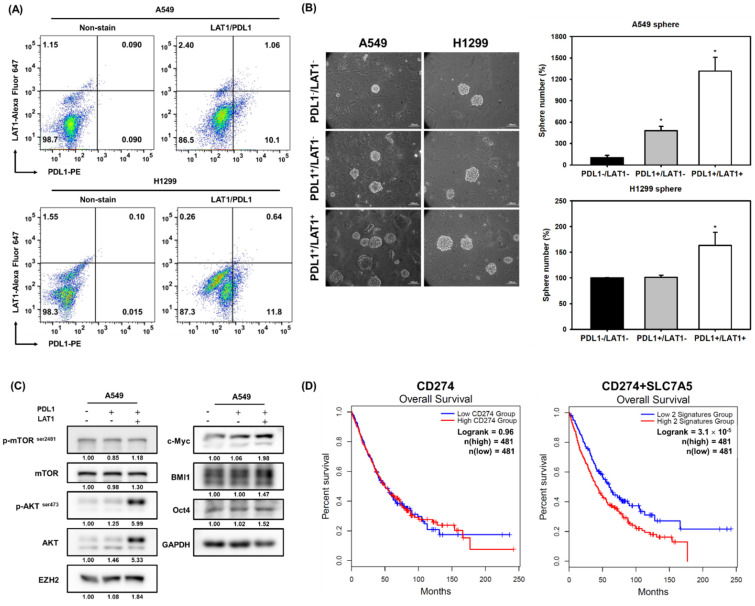
LAT1^+^/PD-L1^+^ enriches CSC activity of NSCLC cells. (**A**) The cell surface expressions of LAT1 and PD-L1 on A549 and H1299 cells were determined by FACS with fluorescence-labeled antibodies. (**B**) Three populations of NSCLC cells (PD-L1^−^/LAT1^−^, PD-L1^+^/LAT1^−^, PD-L1^+^/LAT1^+^) were isolated by FACS cell sorting, and their CSC activities were measured by tumorsphere cultivation. Pictures were presented at 100× magnification and the inserted scale bars indicate 100 μm in length. The data were presented as relative percentages of the tumorsphere number of PD-L1^−^/LAT1^−^ cells. * *p* < 0.05. (**C**) The expressions of cancer stemness proteins (BMI1, c-MYC, EZH2, OCT4), phosphor-mTOR^ser2481^, mTOR, phosphor-Akt^ser473^, and Akt were determined by Western blot. The inserted numbers indicated relative expression levels in comparison to PD-L1^−^/LAT1^−^ cells after normalization to GAPDH. (**D**) The overall survival curves of NSCLC patients in the TCGA database regarding expression levels of CD274 and the two-gene signature of CD274 and SLC7A5 using median level as the cutoff criteria were analyzed using the GEPIA2 webtool [18].

## Data Availability

The RNA Sequencing data could be available from the corresponding author on reasonable request.

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
