# Peer review of "L-Type Amino Acid Transporter 1 Regulates Cancer Stemness and the Expression of Programmed Cell Death 1 Ligand 1 in Lung Cancer Cells"

_ijms, 2021, doi:10.3390/ijms222010955_

Round 1
Reviewer 1 Report
In this manuscript by Liu et al., the authors demonstrate that LAT1 inhibitors can be used as anti-CSC agents and suggest that LAT1 inhibitors can be used in combination with ICIs for the treatment of NSCLC. The manuscript is clearly written, however, prior to publication, the authors need to address the following comments:
1. For Figure 1C, please include a western blot to show the expression of LAT1 in A549, A400 and H1299 cells. This Figure can be included in the Supplementary data.
2. For Figure 3C, are the values shown relative to GAPDH? As can be seen in the western blot, the loading of GAPDH is much lower in shRNA #2 and siSLC7A5 100nM compared to the control. Can the values please represent the expression relative to GAPDH?
3. Could the authors please include scale bars in all their microscopy images.
4. The authors state "the knockdown of LAT1 by RNA interference was found to reduce PD-L1 expression in terms of total protein level (Fig. 4A) and on the cell surface (Fig. 4B) of A400 and H1299 cells by western blot and fluorescence-activated cell sorting (FACS) analysis, respectively". Can the authors please include the total expression of PD-L1 for Figure 4A as stated?
5. The authors refer to Figures 4C and D. This has been mislabelled in the Figure as B. This Figure and its corresponding legend are mislabelled and has to be corrected.
Author Response
First of all, we would like to thank the comments from reviewers that give us the opportunity to strengthen our manuscript. The manuscript has been revised and appropriate changes have been made in accordance with your suggestions. The point-by-point responses are listed below.
- For Figure 1C, please include a western blot to show the expression of LAT1 in A549, A400 and H1299 cells. This Figure can be included in the Supplementary data.
Responses:
We thank the comments from the reviewer. The western blot data of LAT1 expression among three NSCLC cell lines used in this study has been included as Figure S1 as the suggestions by reviewer#1.
- For Figure 3C, are the values shown relative to GAPDH? As can be seen in the western blot, the loading of GAPDH is much lower in shRNA #2 and siSLC7A5 100nM compared to the control. Can the values please represent the expression relative to GAPDH?
Responses:
We apologize the unclear descriptions about our calculations of immunoblot data. All the bands of specific proteins were firstly normalized to their own GAPDH signals followed by setting the values of control groups (i.e. sh-LacZ, negative control siRNA, or 0.1% DMSO according to experimental groups.). We have updated the descriptions in all legends related to immunoblots.
- Could the authors please include scale bars in all their microscopy images.
Responses:
Thank for the comments from the reviewer and the scale bars have been included in all the microscopy images.
- The authors state "the knockdown of LAT1 by RNA interference was found to reduce PD-L1 expression in terms of total protein level (Fig. 4A) and on the cell surface (Fig. 4B) of A400 and H1299 cells by western blot and fluorescence-activated cell sorting (FACS) analysis, respectively". Can the authors please include the total expression of PD-L1 for Figure 4A as stated?
Responses:
We thank the suggestions from the reviewer. The immunoblots of PD-L1 have been included in Figure 4A.
- The authors refer to Figures 4C and D. This has been mislabelled in the Figure as B. This Figure and its corresponding legend are mislabelled and has to be corrected.
Responses:
We apologize for the misleadings of Figure 4 and they have been corrected in both maintext and legends.
Reviewer 2 Report
The manuscript by Liu et al titled “L-type amino acid transporter 1 regulated cancer stemness and the expression of programmed cell death ligand 1 in lung cancer cells” provides data that LAT1 has a role in maintaining a cancer-initiating cell phenotype in NSCLC cells. Previously shown to be upregulated in pemetrexed resistant A549 NSCLC cells, loss of LAT1 through targeted knockdown or pharmacological inhibition of LAT1 activity leads to a reduction in tumor sphere formation in vitro. The loss in tumor spheroid formation correlates with changes in genes associated with tumor initiating cell phenotypes. In addition, loss of LAT1 is stated to lead to changes in mTOR activation. The study then correlates PDL1 levels and LAT1 levels in driving sphere formation in the cell lines and provide TCGA data analysis to show a correlation of high PDL1 and LAT1 levels in reducing NSCLC patient survival.
The introduction needs to be expanded to properly provide background information and rationale for why pemetrexed resistant cells overexpress LAT1. Why was LAT1 expression even considered in this context? It is not clear.
A single assay (sphere formation) to assess tumor initiating activity of a cell line is not sufficient to confidently conclude that tumor initiating activity is altered. In addition, it is not clear what the mechanism of reduced sphere formation – cell cycle arrest, apoptosis, something else - upon modulation of LAT1 is.
Why are H1299 cells used in these experiments when the initial rationale is comparing parental A549 cells to the pemetrexed-resistant A549 A400 cells? How to H1299 cells tie into the pemetrexed-resistant cell model system?
It is not clear why a Spearman Correlation test was done to analyze the data in figure 1E when there are 4 groups to be compared. A Spearmen Correlation is not designed for more than 2 groups. In addition, the presented data do not support the stated conclusion that LAT1 mRNA expression “significantly increased with clinical stage.” In addition, the statistical analyses in the paper require post hoc analyses to correct for multiple comparisons.
What are the extra dotted lines in the overall survival curves presented in figures 1 and 6?
The shRNAs used to knockdown LAT1 in the A400 cells as shown in Figure 2 do not make sense. shRNA 1 has a dramatic effect on sphere formation and LAT1 protein levels are reduced, but mTOR signaling is not affected. Where shRNA 2 affects sphere formation, LAT1 protein levels, and mTOR pathway activation markers. This argues that there is either an off target of the shRNA or the LAT1-dependent sphere formation is mTOR-independent.
In Figure 3, the JPH203 inhibitor does not show a dose-dependent inhibition of mTOR signaling in either A400 cells or in H1299 cells as stated. The 25 uM treatment efficiently inhibits sphere formation but does not block LAT1 activation of mTOR pathway activation markers arguing JPH203 is having off target effects on sphere formation.
Also, in Figure 3E it is not clear why the JPH203 treatment was reduced to 10 uM for the myr-AKT rescue experiment. The sphere reduction is very modest at 10 uM and if the 25 uM does not block LAT1 activity as shown in figure 3B, 10 uM would not block it here making the AKT involvement unclear.
There is not a clear rationale presented to make the jump to examining the effects of LAT1 on PDL1 in section 2.3 of the results.
In Figure 4, the knockdown of LAT1 by shRNA2 is not very good and not similar to that shown in previous figures yet it is shown to have a significant effect on PDL1 surface expression which does not make sense. Again, it argues that the shRNAs used are having off-target effects.
It is not clear why the experiments described in section 2.4 are focused on using the parental A549 cells instead of the pemetrexed-resistant A400 cell line.
There is no control showing the experiments presented in Figure 5 actually have PDL1 knockdown at the protein level. How are the A549 cells in Figure 5A creating so many spheres when Figure 1 shows they are poor at making spheres?
What are the levels used as cut offs for high vs low expression for LAT1 and PDL1 the survival data shown in Figures 1F and 6D?
Author Response
First of all, we would like to thank the comments from reviewers that give us the opportunity to strengthen our manuscript. The manuscript has been revised and appropriate changes have been made in accordance with your suggestions. The point-by-point responses are listed below.
- The introduction needs to be expanded to properly provide background information and rationale for why pemetrexed resistant cells overexpress LAT1. Why was LAT1 expression even considered in this context? It is not clear.
Responses:
We thank the comments from the reviewer. The related paragraph in introduction has been revised by adding one reference that demonstrating the involvement of LAT1 in the resistance to docetaxol in luminal type breast cancer cells (Please see line 80 to line 87 in the revised manuscript).
- A single assay (sphere formation) to assess tumor initiating activity of a cell line is not sufficient to confidently conclude that tumor initiating activity is altered. In addition, it is not clear what the mechanism of reduced sphere formation – cell cycle arrest, apoptosis, something else - upon modulation of LAT1 is.
Responses:
In addition to tumorsphere assay, we also analyzed the expression of proteins involving in the regulation of cancer stemness and results revealed that the knockdown of LAT1 or the treatment of JPH203 caused the downregulation of BMI1 or SOX2 (Fig. 2B and Fig. 3C in this revised manuscript). We believe these data support the involvement of LAT1 in the maintenance of CSCs in NSCLC cells.
- Why are H1299 cells used in these experiments when the initial rationale is comparing parental A549 cells to the pemetrexed-resistant A549 A400 cells? How to H1299 cells tie into the pemetrexed-resistant cell model system?
Responses:
Both of the data in this study and in a previous report from our group (Cancers.) revealed that H1299 displayed a less sensitivity of pemetrexed in compared to A549 (Fig. 1D). Though, we include H1299 NSCLC cells in this study.
- It is not clear why a Spearman Correlation test was done to analyze the data in figure 1E when there are 4 groups to be compared. A Spearmen Correlation is not designed for more than 2 groups. In addition, the presented data do not support the stated conclusion that LAT1 mRNA expression “significantly increased with clinical stage.” In addition, the statistical analyses in the paper require post hoc analyses to correct for multiple comparisons.
Responses:
We thank the critical comments from the reviewer. The stage plot according to SLC7A5 was obtained from TISDB website and we did not know why the Spearman Correlation was used in stage plot. To address the issue raised by the reviewer, we changed the stage plot with the data obtained from GEPIA2 website, which used one-way ANOVA as the statistical analysis method. We agree with the comments from the reviewer in the inappropriate descriptions of the stage plot data and it has been revised as “The lung adenocarcinoma data of the Cancer Genome Atlas (TCGA) database dis-played the significant variations in LAT1 mRNA expression among clinical stages (Fig. 1E)” in this revised manuscript.
- What are the extra dotted lines in the overall survival curves presented in figures 1 and 6?
Responses:
According to the original paper of GEPIA2021 (Nucleic Acids Research, 49: W242-246, 2021) the dotted lines represented the upper and lower 95% confidential intervals. To avoid the confusion it may cause, we remove the dotted lines in Fig. 1F and Fig. 6D.
- The shRNAs used to knockdown LAT1 in the A400 cells as shown in Figure 2 do not make sense. shRNA 1 has a dramatic effect on sphere formation and LAT1 protein levels are reduced, but mTOR signaling is not affected. Where shRNA 2 affects sphere formation, LAT1 protein levels, and mTOR pathway activation markers. This argues that there is either an off target of the shRNA or the LAT1-dependent sphere formation is mTOR-independent.
Responses:
We agree with the comments from the reviewer. Actually, we have tried to transfect the siRNA oligos into A400 cells but it did not work well because of the low transfection efficiency. Nevertheless, the data of shRNAs in A400 and siRNA in H1299 cells supported the hypothesis that LAT1 involves in CSC maintenance and PD-L1 expression. In the experiment of myr-Akt overexpression, we found that the expression of myr-Akt did not overcome the suppressive effect of JPH203 in tumorsphere formation at higher concentration of 25 mM, which suggests the existence of Akt/mTOR independent pathway in LAT1 mediated regulation of self-renewal capability as the reviewer mentioned. To improve our manuscript, we added the data of 25 mM JPH203 in myr-Akt overexpression experiment in Figure 3F and have made some discussions to this observeation in the Discussion section (Line 303 to line 308).
- In Figure 3, the JPH203 inhibitor does not show a dose-dependent inhibition of mTOR signaling in either A400 cells or in H1299 cells as stated. The 25 uM treatment efficiently inhibits sphere formation but does not block LAT1 activation of mTOR pathway activation markers arguing JPH203 is having off target effects on sphere formation. Also, in Figure 3E it is not clear why the JPH203 treatment was reduced to 10 uM for the myr-AKT rescue experiment. The sphere reduction is very modest at 10 uM and if the 25 uM does not block LAT1 activity as shown in figure 3B, 10 uM would not block it here making the AKT involvement unclear.
Responses:
In the experiment of myr-Akt overexpression, we found that the expression of myr-Akt did not overcome the suppressive effect of JPH203 in tumorsphere formation at higher concentration of 25 mM, which suggests the existence of Akt/mTOR independent pathway in LAT1 mediated regulation of self-renewal capability as the reviewer mentioned. To improve our manuscript, we added the data of 25 mM JPH203 in myr-Akt overexpression experiment in Figure 3F and have made some discussions to this observeation in the Discussion section (Line 303 to line 308).
- There is not a clear rationale presented to make the jump to examining the effects of LAT1 on PDL1 in section 2.3 of the results.
Responses:
We thank the comments from the reviewer. Due to the reduction of Akt/mTOR activation in LAT1 knockdown or JPH203 treatment NSCLC cells and the involvement og Akt/mTOR pathway in PD-L1 expression in cancer cells, we decided to understand the effect of LAT1 in PD-L1 expression. We have strengthen the descriptions about the rationale for investigating the effect of LAT1 in PD-L1 expression in line 183 to line 187.
- In Figure 4, the knockdown of LAT1 by shRNA2 is not very good and not similar to that shown in previous figures yet it is shown to have a significant effect on PDL1 surface expression which does not make sense. Again, it argues that the shRNAs used are having off-target effects.
Responses:
Although the results of sh-SLC7A5#2 might raise the concern of off-target effect, the data of sh-SLC7A5#1 in A400 cells and si-SLC7A5 in H1299 cells, as well as the data of JPH203 treatment in A400 and H1299 cells (Fig. 4C and Fig, 4D) support the hypothesis in the positive regulation role of LAT1 in PD-L1 expression. We hope these data could persuade the reviewer to accept our explanations.
- It is not clear why the experiments described in section 2.4 are focused on using the parental A549 cells instead of the pemetrexed-resistant A400 cell line.
Responses:
We thank the comments from the reviewer. Even though this experiment used A549 cells instead of A400 cells, the results also support our hypothesis of the involvement of PD-L1 in CSC activity of NSCLC cells.
- There is no control showing the experiments presented in Figure 5 actually have PDL1 knockdown at the protein level. How are the A549 cells in Figure 5A creating so many spheres when Figure 1 shows they are poor at making spheres?
Responses:
We apologize for the lack of PD-L1 expression data in Figure 5 and they have been added in this revised manuscript. Due to the low efficiency of tumorsphere formation of A549 cells, we seeded 1´104 cells/well initially in Figure 5 and this information has been included in Materials and Methods section (section 4.2, line 346 to line 347).
- What are the levels used as cut offs for high vs low expression for LAT1 and PDL1 the survival data shown in Figures 1F and 6D?
Responses:
The cutoff expression levels of LAT1 or CD274 in Figure 1F and Figure 6D were median. The information have been included in the legends.
Round 2
Reviewer 2 Report
The authors have greatly improved the manuscript with the changes made. There are 2 minor issues that need to be addressed by adding to the text. My comments were not clear enough in the initial review. I have included the original comment, response, and my clarification below (numbering is based on original review).
3. Why are H1299 cells used in these experiments when the initial rationale is comparing parental A549 cells to the pemetrexed-resistant A549 A400 cells? How to H1299 cells tie into the pemetrexed-resistant cell model system?
Responses:
Both of the data in this study and in a previous report from our group (Cancers.) revealed that H1299 displayed a less sensitivity of pemetrexed in compared to A549 (Fig. 1D). Though, we include H1299 NSCLC cells in this study.
- This explanation needs to be included in the manuscript. The problem is that the use of the H1299 as a control cell line is not clearly explained in the text and remains confusing for the reader. It is good to use the H1299 cells as a control cell line for the experiments, but it should be explicitly stated in the manuscript and the reference to the Cancers paper noted.
10. It is not clear why the experiments described in section 2.4 are focused on using the parental A549 cells instead of the pemetrexed-resistant A400 cell line.
Responses:
We thank the comments from the reviewer. Even though this experiment used A549 cells instead of A400 cells, the results also support our hypothesis of the involvement of PD-L1 in CSC activity of NSCLC cells.
- As above, the reasoning for the use of the parental A549 cells needs to be stated in the manuscript to avoid reader confusion. Simply switching experimental model systems without explanation makes the logical flow of the experiments confusing.
Author Response
First of all, we would like to thank the general positive feedbacks from reviewers and the suggestions to improve our manuscript. The manuscript has been revised and appropriate changes have been made in accordance with your suggestions.
- This explanation needs to be included in the manuscript. The problem is that the use of the H1299 as a control cell line is not clearly explained in the text and remains confusing for the reader. It is good to use the H1299 cells as a control cell line for the experiments, but it should be explicitly stated in the manuscript and the reference to the Cancers paper noted.
Responses:
We thank the comments from the reviewer and the following descriptions have been added to the first paragraph of Results section as “ We previously found that the sensitivity of H1299 cells to pemetrexed was between A549 cells and A400 cells [17]. Thus, the H1299 cells were included to investigate the relationship between LAT1 and pemetrexed sensitivity. With quantitative reverse transcription-polymerase chain reaction (qRT-PCR) and fluorescence activated cell sorter (FACS) analyses, we…..” (line 101 to line 105).
- As above, the reasoning for the use of the parental A549 cells needs to be stated in the manuscript to avoid reader confusion. Simply switching experimental model systems without explanation makes the logical flow of the experiments confusing.
Responses:
Here we would like to investigate the general impact of expression patterns of cell surface LAT1/PDL1 on the CSC activity in NSCLC cells, thus we used A549 and H1299 cells, rather than pemetrexed resistant A400 cells. The rationale for using A549 and H1299 cells in this part of experiments has been added as follows:” Due to the observations of the positive roles of LAT1 or PD-L1 expression in CSC activity in NSCLC cells, we next used the parental A549 or H1299 cells, rather than the pemetrexed resistant A400 cells, to understand the general impact of the expression pat-terns of cell surface LAT1/PD-L1 on the CSC activity of NSCLC cells.” (line 247 to line 250).